# A mRNA Vaccine for Crimean–Congo Hemorrhagic Fever Virus Expressing Non-Fusion GnGc Using NSm Linker Elicits Unexpected Immune Responses in Mice

**DOI:** 10.3390/v16030378

**Published:** 2024-02-28

**Authors:** Tong Chen, Zhe Ding, Xuejie Li, Yingwen Li, Jiaming Lan, Gary Wong

**Affiliations:** 1Viral Hemorrhagic Fever Research Unit, Chinese Academy of Sciences (CAS) Key Laboratory of Molecular Virology & Immunology, Shanghai Institute of Immunity and Infection (Formerly Institut Pasteur of Shanghai), Chinese Academy of Sciences, Shanghai 200031, Chinaxjli@siii.cas.cn (X.L.);; 2University of Chinese Academy of Sciences, Beijing 100049, China

**Keywords:** Crimean–Congo hemorrhagic fever virus, NSm, Gn, Gc, mRNA vaccine, LNPs

## Abstract

Crimean–Congo hemorrhagic fever (CCHF), caused by Crimean–Congo Hemorrhagic virus (CCHFV), is listed in the World Health Organization’s list of priority diseases. The high fatality rate in humans, the widespread distribution of CCHFV, and the lack of approved specific vaccines are the primary concerns regarding this disease. We used microfluidic technology to optimize the mRNA vaccine delivery system and demonstrated that vaccination with nucleoside-modified CCHFV mRNA vaccines encoding GnNSmGc (vLMs), Gn (vLMn), or Gc (vLMc) induced different immune responses. We found that both T-cell and B-cell immune responses induced by vLMc were better than those induced by vLMn. Interestingly, immune responses were found to be lower for vLMs, which employed NSm to link Gn and Gc for non-fusion expression, compared to those for vLMc. In conclusion, our results indicated that NSm could be a factor that leads to decreased specific immune responses in the host and should be avoided in the development of CCHFV vaccine antigens.

## 1. Introduction

Crimean–Congo hemorrhagic fever (CCHF) disease is usually caused by bites from ticks carrying Crimean–Congo hemorrhagic fever virus (CCHFV) or direct contact with the blood or tissue of viremia-phase livestock [1]. Infected patients usually present symptoms of myalgia, fever, nausea, vomiting, and diarrhea in the pre-hemorrhagic period and hematemesis, melena, and somnolence in the hemorrhagic period, with a case fatality rate of around 30% [2,3,4]. CCHFV has been reported in Asia, Africa, the Middle East, and Europe [5] and is divided into the Asia-1, Asia-2, Africa-1, Africa-2, Africa-3, Europe-1, and Europe-2 clades [6]. Although *Hyalomma* ticks are the major vectors carrying CCHFV, researchers have found that other tick species can also potentially carry CCHFV [7,8]. With climate change, the geographical range of ticks carrying CCHFV may expand [5], resulting in an increase in the outbreak risk for CCHF. Specific antiviral drugs and vaccines are not yet available for CCHF, and the World Health Organization (WHO) has listed this disease as a priority for research and development (https://www.who.int/news-room/events/detail/2018/02/06/default-calendar/2018-annual-review-of-diseases-prioritized-under-the-research-anddevelopment-blueprint, (accessed on 7 February 2018)), highlighting the importance of research advances in this field.

CCHFV belongs to the *Nairoviridae* family of the *Bunyavirales* order [2]. The genome consists of three single-stranded, negative-sense RNA segments: (1) small (S); (2) medium (M); and (3) large (L). The S segment encodes nucleocapsid (NP) and non-structural (NSs) proteins, while the M segment encodes the glycoprotein precursor (GPC), containing the mucin-like domain (MLD), GP38, N-terminus glycoprotein (Gn), non-structural protein (NSm), and C-terminus glycoprotein (Gc). The L segment, on the other hand, encodes the RNA-dependent RNA polymerase (RdRp). The major immunogen for the development of CCHFV vaccines is the viral glycoprotein, which plays a role in the attachment of the virus particle to the host cell and elicits T- and B-cell responses in addition to neutralizing antibodies [9,10]. Other antigens have also been explored for their potential to stimulate more robust immune responses. To date, vaccines using NP as an immunogen successfully induce T-cell immune responses and antibodies against CCHFV, with efficacious protection observed after challenge in animal models [11,12,13]. However, not all NP vaccines were found to protect 100% of immunized animals, and none of them can induce neutralizing antibodies [13,14]. Furthermore, GP38 plays a key role to protect 100% of mice from CCHFV challenge when expressed with the inactivated rhabdovirus-based vaccine platform [15].

Vaccines targeting different formulations of the viral glycoprotein against CCHFV have also been explored as candidates. Interestingly, it was found that immune response levels and protection efficiency were not ideal when using DNA or mRNA vaccine platforms encoding the GPC as the immunogen [10,16,17]. Conversely, a CCHFV mRNA vaccine encoding GcGn protected 100% of IFNAR^−/−^ mice from a lethal CCHFV challenge [9]. Previously, a Rift Valley fever virus (RVFV) DNA vaccine candidate encoding NSmGnGc failed to protect IFNAR^−/−^ mice from RVFV challenge, but GnGc protected 100% of IFNAR^−/−^ mice [18]. Considering that both CCHFV and RVFV belong to the order *Bunyavirales*, we speculated that the addition of NSm to CCHFV may possibly play a similar role in disrupting the immune effects of candidate glycoprotein-based CCHFV nucleic acid vaccines.

In this study, we designed three nucleoside-modified mRNA vaccines, which encode for full-length Gn (vLMn), full-length Gc (vLMc), and GnNSmGc (vLMs) of the YL16070 strain of CCHFV. To deliver the various mRNA vaccine candidates into mice, we established and optimized a lipid nanoparticles (LNPs) delivery system by microfluidics, using enhanced green fluorescent protein (eGFP) and luciferase (Luc) as reporter genes. Afterwards, we assessed these CCHFV mRNA vaccines for specific T- and B-cell immune responses after primary and boost immunizations. We found that vLMc is superior to vLMn in terms of immune response levels, which is consistent with previously published results from the subunit vaccine platform expressing Gn or Gc [19,20]. Immunization with vLMs was found to result in decreased immunity compared to that with vLMc, supporting the notion that NSm could be a factor that plays a role in the decreased immunity of the CCHFV nucleic acid vaccines. Thus, suggesting the exclusion of NSm in future vaccine candidates may be important in the generation of more immunogenic nucleic acid vaccines.

## 2. Materials and Methods

### 2.1. Ethics Statement

In this study, we used 6- to 8-week-old immunocompetent female C57BL/6J and BALB/c mice. All research involving live animals was conducted according to the ethical guidelines of the Shanghai Institute of Immunity and Infection, Chinese Academy of Sciences (Ethics No. A2022048). The mice were kept under specific pathogen-free conditions, observed twice daily, and fed food and water ad libitum. Immunizations were performed intramuscularly (i.m.), and mice were euthanized by an overdose of inhalational isoflurane gas, followed by cervical dislocation after the end of the observation period.

### 2.2. Plasmid Construction

The sequence of the M segment of CCHFV YL16070 strain was downloaded from Genbank (AQX34599.1) and synthesized commercially (GenScript, Nanjing, China). The mRNA template plasmid pGEM-3Zf (+)-Model encoding 3′UTRN, 5′UTR (Appendix A) and Poly(A) was synthesized commercially (GenScript, Nanjing, China). We designed primers and amplified eGFP, Luc, Gn, GnHis, Gc, GcHis, GnNSmGc, and GnNSmGcHis, following the protocol of PrimerStar (R045A, Takara, Beijing, China) for the construction of various plasmids pGEM-3Zf (+) mt-eGFP/Luc/Gn/GnHis/Gc/GcHis/GnNSmGc/GnNSmGcHis. The PCR products were cut, gel-purified, and cloned into the pGEM-3Zf (+)-Model by ClonExpress^®^ Ultra One Step Cloning Kit (C115, Vazyme, Nanjing, China). Finally, the various plasmids pGEM-3Zf (+) mt-eGFP/Luc/Gn/Gc/GnNSmGc/GnHis/GcHis/GnNSmGcHis were confirmed by Sanger sequencing (Biosune, Shanghai, China).

### 2.3. mRNA Synthesis

mRNA was produced and purified by T7 High Yield RNA Synthesis Kit (E2040S, NEB, Beijing, China), Vaccinia Capping System (M2080S, NEB, Beijing, China), pseudouridine (ψ) (N-1019, TriLink, Shenzhen, China), mRNA Cap 2′-O-Methyltransferase (M0366S, NEB, Beijing, China), DNase I (M0303S, NEB, Beijing, China), and Monarch^®^ RNA Cleanup Kit (T2050L, NEB, Beijing, China), as follows: (a) Linearize 8 μg of pGEM-3Zf (+) mt-eGFP/Luc/Gn/GnHis/Gc/GcHis/GnNSmGc/GnNSmGcHis plasmid with 4 μL NsiI-HF enzyme (R3127, NEB, Beijing, China) overnight at 37 °C. (b) Verify above plasmids for complete linearization by gel electrophoresis, before purification with Cycle-Pure Kit (D6492, Omega, Beijing, China). (c) Formulate reaction system (Table 1) in tubes and perform in vitro transcription (IVT) for 10 h at 37 °C. Afterwards, the RNA products were treated with 2 μL DNase I (M0303S, NEB, Beijing, China) for 30 min at 37 °C, and the products were purified by Monarch^®^ RNA Cleanup Kit. (d) Following the Vaccinia Capping System (M2080S, NEB, Beijing, China) and mRNA Cap 2′-O-Methyltransferase (M0366S, NEB, Beijing, China) protocol, Cap 1-mRNA- eGFP/Luc/Gn//GnHis/Gc/GcHis/GnNSmGc/GnNSmGcHis was generated, and after purification with Monarch^®^ RNA Cleanup Kit, 1 μg/μL per tube was aliquoted and stored in −80 °C for future use.

### 2.4. eGFP/Luc Expression Studies by Fluorescence Microscopy

The 293T cells were seeded in 6-well plates at 2 × 10^6^ cells/well with 10% fetal bovine serum (FBS) DMEM. Sixteen hours later, the cells were transfected with Cap1-mRNA-eGFP or control mRNA (2.5 μg/well) using the Lipofectamine^TM^ 2000 Reagent (Invitrogen, Carlsbad, CA, USA). At 6 h, 12 h, 24 h, 48 h, 72 h, and 96 h, eGFP expression was observed under an inverted fluorescence microscope (Olympus, Tokyo, Japan).

The 293T cells were resuspended with FBS-free DMEM, then seeded in 96-well plates at 1 × 10^5^ cells/well. Sixteen hours later, the cells were transfected with Cap1-mRNA-Luc or control mRNA (200 ng/well) using the Lipofectamine^TM^ 2000 Reagent. Luc activity was detected at 6 h, 12 h, 24 h, 48 h, and 72 h by a microplate reader (Bio Tek, Winooski, VT, USA).

### 2.5. Protein Expression Studies by Western Blot and Immunofluorescence

The 293T cells were seeded in 6-well plates at 2 × 10^6^ cells/well with 10% FBS DMEM. Sixteen hours later, the cells were transfected with Cap1-mRNA-Gn/Gc/GnNSmGc/GnHis/GcHis/GnNSmGcHis or control mRNA Cap1-mRNA (2.5 μg/well) using the Lipofectamine^TM^ 2000 Reagent. After 24 h, GnHis/GcHis/GnNSmGcHis expression was detected by Western Blot with mouse anti-His-tag monoclonal antibody (AE003, ABclonal, Wuhan, China).

At the same time, the supernatant of Gn/Gc/GnNSmGc expression group was removed, and the cells were washed twice with PBS. Afterwards, 293T cells were fixed with 4% paraformaldehyde (Solarbio, Beijing, China) for 20 min and washed twice with PBS. The 293T cells were permeabilized with 0.1% Triton-X-100 (Sigma, Shanghai, China) for 20 min and washed twice with PBS, prior to blocking with 5% FBS PBS for 30 min. After two washes with PBS, Gn/Gc serum (a generous gift provided by the Wuhan Institute of Virology, CAS, Wuhan, China) was diluted 1:100 with PBS and incubated at 25 °C for 2 h before washing three times with PBS. Anti-mouse Alexa Fluor^®^ 555 Molecular Probes (4409, Cell Signaling Technology, Shanghai, China), diluted 1:1000 with PBS, were incubated for 1 h in darkness. After three washes with PBS, DAPI (4083, Cell Signaling Technology), diluted 1:10,000 with PBS, was incubated for 8 min in darkness. After two washes with PBS and one wash with nuclease-free water, the signals were detected by an inverted fluorescence microscope.

### 2.6. Establishment of the mRNA Vaccine Delivery System

We used an orthogonal experimental design based on the classical formulation of LNPs [21] to optimize an mRNA delivery system. DLin-MC3-DMA (O02006, AVT, Shanghai, China) was combined into an ethanol phase with cholesterol (C804519, Shanghai Macklin Biochemical, Shanghai, China), DOPE (850725P, Sigma, Shanghai, China), and 14:0 PEG2000 (880150P, Sigma, Shanghai, China) at different molar ratios with a total volume of 100 μL. Meanwhile, the separate aqueous phase was prepared using 30 μg of Cap1-mRNA-eGFP and different concentrations of citrate buffer with different pH values with a total volume of 300 μL (Table 2 and Table 3). The flow velocity ratio of the ethanol phase and the aqueous phase is 1:3. All LNPs-Cap1-mRNA products were generated by microfluidics chips mixture. LNPs-Cap1-mRNA was dialyzed against PBS for 2 h before sterile filtration through 0.22 μm filters. All operations were carried out under RNase-free conditions.

Afterwards, we built on the results of the orthogonal experimental design by further screening for the optimal LNP delivery system using a complete randomized design (Table 4), where the concentration of citrate buffer was 50 mM. All experimental environments are the same as the processes described above for LNP manufacture.

For detecting encapsulation efficiency, we conducted a gel retardation assay [22]. Briefly, the electrophoresis tank was soaked with 1% DEPC water overnight to remove RNase prior to running the gel. The electrophoresis tank was rinsed with 1 × TBE buffer and a 1.2% agarose gel was prepared in 1 × TBE buffer. The LNP–mRNA samples collected by orthogonal experimental design and complete randomized design were run on the gel for 30 min at 100 V. For determining the physical characteristics of LNPs-Cap1-mRNA, the radius of LNPs was detected using DynaPro NanoStar (Wyatt Technology, Santa Barbara, CA, USA) for uniformity.

For verifying whether LNPs-mRNA-eGFP could be translated in eukaryotic cells, 293T cells were seeded in 6-well plates at 2 × 10^6^ cells/well. Sixteen hours later, the cells were transfected with LNPs-Cap1-mRNA-eGFP or control LNPs-Cap1-mRNA generated by microfluidics technology. After 24 h, eGFP expression was detected using an inverted fluorescence microscope.

### 2.7. Cytotoxicity Detection of LNPs In Vitro

The 293T cells were seeded in 96-well plates at 10^5^ cells/well with FBS-free DMEM. Sixteen hours later, the cells were incubated with FBS-free DMEM with 0.5 mM, 1 mM, 1.5 mM, 2.0 mM, or 2.5 mM LNPs. After 24 h, 10 μL of CCK-8 (A311-01, Vazyme, Nanjing, China) solution was added into each sample well. After 2 h, the observed color change of the media was detected by a microplate reader.

### 2.8. Luc Expression In Vivo

For the detection of in vivo expression of the optimized mRNA vaccine delivery system, female BALB/c mice (*n* = 3) were administrated with 10 μg of LNPs-Cap1-mRNA-Luc, or via i.m., using the same volume of PBS. After 6 h, 12 h, 24 h, and 48 h post-inoculation, the mice were injected intraperitoneally (i.p.) with D-Luciferin potassium (HY-12591B, MedChemExpress LLC, Shanghai, China). After 15 min, fluorescence signals were collected using IVIS Spectrum instrument (PerkinElmer, Waltham, MA, USA) for 60 s.

### 2.9. Immunization Schedule of CCHFV mRNA Vaccines

For two-dose vaccination, groups of C57BL/6J mice (*n* = 5) were immunized i.m. with 10 μg of either mock (LNPs-Cap1-mRNA-Luc), vLMn, vLMc, or vLMs vaccines and boosted with an identical injection via i.m. on day 21. Sera were collected at days 7, 14, 28, and 35 for the detection of Gn- or Gc-specific IgG antibody responses as described below. Spleen tissues were collected at days 7 and 21 for the evaluation of cellular immune responses by Enzyme Linked Immunospot assay (ELISPOT) as described below. The schematic of the timeline is shown in Figure 1.

### 2.10. CCHFV Gn and Gc Protein Expression in Cell Lysates

Based on a previously published method [23], 10^7^ 293T cells were seeded in 15 cm dishes. Sixteen hours later, 30 μg of pcDNA3.1-Gn plasmid or pcDNA3.1-Gc plasmid were transfected into cells using the Lipofectamine^TM^ 2000 Reagent following the manufacturer’s instructions. The cells were incubated with 5% CO_2_ at 37 °C for 24 h. Afterwards, the supernatant was removed, then the cells were washed twice with PBS, digested with Trypsin-EDTA, resuspended, and moved into 15 mL tubes with 3 ml PBS. The tubes were placed in a mixture of ice and water, and the cells were lysed by sonication and were centrifuged at 12,000× *g* at 4 °C for 15 min. The supernatant was collected, and the total protein was detected (including Gn or Gc) by the Protein Content Assay Kit (BC3180, Solarbio) following the manufacturer’s instructions. The protein was stored at −80 °C until future use.

### 2.11. Enzyme Linked Immunosorbent Assay (ELISA) on CCHFV Gn/Gc Cell Lysates

Total protein (including Gn or Gc) was pre-incubated in 96-well ELISA plates (300 ng/well) at 4 °C overnight. The plates were washed twice with PBS for 2 min on a shaker prior to blocking with 5% skim milk diluted in PBS-Tween (PBST) at 25 °C for 1 h, before washing again as described above. Serum samples collected from mice were serially diluted in a 2-fold gradient with 5% skim milk diluted in PBST, before incubating on the prepared ELISA plates at 37 °C for 2 h. Afterwards, plates were washed with PBST four times for 2 min each on the shaker. The secondary antibodies Goat pAb to Ms IgG HRP (ab6789, abcam) or Goat pAb to Ms IgM HRP (ab97230, abcam) were diluted 1:30,000 with 5% skim milk diluted in PBST, before incubating on the ELISA plates at 37 °C for 1 h. The plates were washed with PBST five times for 2 min each on a shaker. Lastly, the supernatant was removed, and the substrate solution was added. After 12 min, ELISA stop solution (C1058, Solarbio) was added, and the signal was read by a microplate reader (Bio Tek) at 450 nm.

### 2.12. ELISPOT Assay

T-cell immune responses in immunized mice were evaluated via pre-coated interferon-γ (IFN-γ) and interleukin-4 (IL-4) ELISPOT kit (MabTech, Nacka Strand, Sweden) following the manufacturer’s instructions. Briefly, the plates were activated using PBS, blocked using RPMI 1640 medium (Gibco) containing 10% FBS, and incubated for 30 min. Afterwards, immunized mouse splenocytes were plated at 5 × 10^5^ cells/well, with peptide pools spanning the entire CCHFV GPC protein. PMA + Ionomycin (2030421, DAKEWE) was used as the positive control, and RPMI 1640 medium as the negative control. The splenocytes were incubated at 5% CO_2_ at 37 °C for 24 h, and plates were washed with PBS, incubated with IFN-γ or IL-4 antibody, and diluted 1:1000 in PBS at 37 °C for 2 h. Lastly, following the addition of streptavidin-ALP antibody diluted 1:1000 with PBS including 0.5% FBS and incubated at 37 °C for 1 h, the plates were washed with PBS five times for 2 min each on a shaker. The air-dried plates were read, and the spots were counted using the ImmunoSpot reader (CTL).

### 2.13. Tools and Data Analysis

SnapGene version 4.2.1 was used for primer and plasmid design. The data were produced in a graphical format by GraphPad Prism version 9.4.1 and Origin version 2019b 32 Bit. The data were analyzed by two-way ANOVA with multiple comparison test.

## 3. Results

### 3.1. Generation of Candidate CCHFV Vaccines Using the mRNA Plasmid Backbone

The schematics for the plasmid to generate various mRNA are shown in Figure 2A. The target antigen (Figure 2A) encoded eGFP and Luc to detect if the mRNA vaccine platform was functional (Figure 2B). Afterwards, immunogen (Appendix A) genes were optimized for humanization by encoding Gn and GnHis, Gc and GcHis, as well as GnNSmGc and GnNSmGcHis, respectively, for the development of CCHFV mRNA vaccines (Figure 2B). In the latter immunogen, NSm will be cut by the enzyme SKI-1/S1P at the RKLL site, and the enzyme SKI-1/S1P-like at the RKPL site, respectively, so that Gn and Gc will be secreted and transported into the Golgi, resulting in their accumulation in the Golgi compartment along with NSm, thereby promoting Gn and Gc maturation and transport [24,25,26,27].

To determine antigen expression, Cap1-mRNA-eGFP and Cap1-mRNA-Luc were synthesized by IVT and the capping kit. Through the transfection of Cap1-mRNA-eGFP and the positive control (PC) plasmid in 293T cells, the results indicated that eGFP was successfully expressed by Cap1-mRNA at 6 h, and by the PC plasmid at 24 h (Figure 3A). Furthermore, after the detection of Cap1-mRNA-Luc translation in 293T cells, it was clearly observed that the efficiency of modified mRNA was better than that of unmodified mRNA. Meanwhile, Luc activity peaked at 24 h and was found to be sustained at least to 72 h (Figure 3B). Additionally, GnHis (35 KDa) and GcHis (72 KDa) proteins were successfully expressed by relevant Cap1-mRNA in 293T cells (Figure 3C). As expected, GnNSmGcHis was digested to individual Gn and GcHis proteins by SKI-1/S1P and SKI-1/S1P-like enzymes (Figure 3C). Meanwhile, the results of immunofluorescence assays further demonstrated the expression of CCHFV glycoprotein Gn by both Cap1-mRNA-Gn (Figure 3(Di)) and Cap1-mRNA-GnNSmGc in 293T cells (Figure 3(Dii)). Interestingly, compared to Gc expressed by Cap1-mRNA-Gc, Gc expressed by Cap1-mRNA-GnNSmGc was weaker, which was consistent with the immunoblotting results (Figure 3C,(Div,Dv)).

### 3.2. Generation of Lipid Nanoparticles for mRNA Vaccine Delivery

LNPs have been shown to efficiently deliver mRNA vaccines [28]. The mRNA vaccine delivery system, which was based on classical formulation [21], was optimized by microfluidics technology in this study. For determining the best conditions for the delivery of LNPs containing mRNA into animals, the methods of orthogonal experimental design and complete randomized design allowed for the testing of parameters systematically under the evaluation of the particle radius of LNPs-Cap1-mRNA, encapsulation efficiency, and expression in vitro. The characterization of LNPs-Cap1-mRNA is shown in Figure 4. According to panels (i–iii), the size consistency of LNPs manufactured via methods 1, 2, and 3 in Table 3 was good, in which the particle diameters ranged between 76 and 136 nm. The distribution of particle size was dispersed gradually. However, panels (iv-vi) showed that the LNP size manufactured by methods 4, 5, and 6 in Table 3 was unsatisfactory. Although the particle diameters ranged between 85 and 163 nm, the distribution of particle size was not dispersed gradually. Meanwhile, according to gel retardation assay, we evaluated encapsulation efficiency, and the results indicated that Cap1-mRNA could be almost completely encapsulated using methods 1, 2, and 3 but could not be completely encapsulated using methods 4, 5, and 6 (Figure 5A). To further confirm the most optimal formulation using these methods, we analyzed the expression intensity after generating LNPs-Cap1-mRNA-eGFP by methods 1–6 via transfection in 293T cells, and the results showed that method 1 had a good potential to deliver mRNA and that this formulation could deliver Cap1-mRNA-eGFP to cells with the highest levels of subsequent translation (Figure 5B). Therefore, we chose LNPs generated by method 1 to encapsulate the target mRNA as candidate vaccines.

For ensuring that the mRNA delivery system is safe to animals, we performed a cytotoxicity assay to ensure that the LNPs could be tolerated by cells, and the results showed that method 1 formulation maintained 96.1% cell viability when a concentration of 1.5 mM LNPs was used. Thus, we inferred that using an LNP concentration of 3.32 mM would reach the half-maximal inhibitory concentration (IC_50_) (Figure 5C). Next, we evaluated the in vivo expression efficacy of LNPs-Cap1-mRNA-Luc in mice. To visualize expression time and intensity, we used method 1 to encapsulate Cap1-mRNA-Luc, injected the mice by i.m. with the LNP formulations, and subjected the mice to live bioluminescence analysis. Six hours after administration, the presence of Luc was observed at the injection site in BALB/c mice. The Luc signals peaked at 12 h after infection and became undetectable at around 48 h (Figure 5D), demonstrating that the formulation derived from method 1 could successfully deliver mRNA to mice in vivo. Furthermore, no LNPs-Cap1-mRNA-Luc were observed to spread to the liver and kidneys, and the LNPs remained localized to the injection site at 12 h after infection.

### 3.3. CCHFV mRNA Vaccine Induced B-Cell Immune Responses

To investigate whether the vaccine candidates could induce specific humoral immune responses, C57BL/6J mice were divided into four groups (*n* = 5 per group), and each animal was injected with 10 μg of vLMn, vLMc, vLMs, and mock vaccine (LNPs-Cap1-mRNA-Luc) via the i.m. route, following the above schedule (Figure 1). The serum was collected at days 7, 14, 28, and 35. To test if vaccination could generate antibodies against Gn or Gc, the serum samples were measured by ELISA. We generated Gn or Gc via plasmid transfection in 293T cells and used cell lysates of the total protein as a capture antigen to detect a specific antibody.

The individual serum samples were serially diluted 2-fold from 1:25 to 1:200, and specific antibodies were detected. No antibodies were detected in the mock group (Figure 6A,B). Low levels of IgG antibodies of titer (1:25) targeting Gn were observed in the vLMn group after boost immunization (Figure 6A), whereas IgG antibodies against Gn were observed only in one mouse from the vLMs group after boost immunization (Figure 6A). Interestingly, the level of titer (1:25) IgG antibodies targeting Gc induced by the vLMc vaccine after initial immunization (Figure 6B) was comparable to that of IgG antibodies targeting Gn in the vLMn group after boost immunization (Figure 6A). The titer of IgG antibodies targeting Gc then reached 1:100 after a vLMc vaccine boost (Figure 6B). The titer of IgG antibodies targeting Gc could be detected as (1:25) after vLM boost immunization (Figure 6B), which was different to that of those targeting Gn in the vLMs group. No IgM antibody was detected in any of the groups (Appendix A). These results indicated that the titers of Gn- and Gc-specific IgG antibodies were lower when the immunogen included NSm.

### 3.4. CCHFV mRNA Vaccine Induced T Cell Immune Responses

To evaluate cellular immune responses from immunized mice, splenocytes were collected at 7 days after each immunization. Fresh splenocytes from four groups (*n* = 5 per group) were pooled and incubated with peptides based on CCHFV pools spanning GPC. We conducted IFN-γ and IL-4 ELISPOT assays to characterize whether T-cell immune responses were skewed towards Th1 or Th2. Interestingly, neither initial nor boost immunization of vLMn induced IFN-γ in response to any peptide stimulation (Figure 7A,B). Initial immunization of vLMc generated detectable T-cell responses (ranging from 12 to 844 SFCs/10^6^ splenocytes) to the Gc1 peptide (Figure 7A), and it became stronger (ranging from 286 to 648 SFCs/10^6^ splenocytes) after boost immunization (Figure 7B). However, IFN-γ-secreting cells were not detected after initial immunization with vLMs (Figure 7A). Even after boost immunization, the T-cell immune response of the vLM group to the Gc1 peptide was comparatively lower (ranging from 70 to 232 SFCs/10^6^ splenocytes) than that of the vLMc group(Figure 7B). Interestingly, non-specific responses of IL-4 were detected for the three vaccine candidates with the ELISPOT assay (Appendix A).

## 4. Discussion

There are currently several bottlenecks to the development and approval of an effective vaccine against CCHFV: (1) To date, the small animal model available for screening candidates is the interferon knockout or interferon-blocked mice [29], which prevents a robust evaluation of vaccine-induced responses; (2) GP is highly divergent between different CCHFV clades, which poses issues for the design of a universal CCHFV vaccine, whereas NP is more conserved between CCHFV isolates but cannot induce neutralizing antibodies [30]; (3) It is currently unclear whether the stimulation of neutralizing antibodies against CCHFV is required for protection, as some studies show neutralizing antibodies are not required for protection from lethal mice infection [23,31], whereas other studies are inconclusive on the usefulness of neutralizing antibodies in protecting humans from CCHF [32,33]; (4) The same antigen expressed via different vaccine platforms confers different levels of efficacy [16,34]. Additionally, some unconventional antigens complicate the development of CCHFV vaccines. Past studies have shown that GP38, a protein from the CCHFV M segment, could be located on the virion surface, and they have demonstrated that it alone could induce specific immune responses and play an important role in protecting the host against CCHFV infection [15,23,35]. Interestingly, GP38 was able to improve protection efficiency in the inactivated rhabdovirus-based CCHFV-M vaccine. However, protection efficiency was not very ideal when using the M segment of CCHFV, including NSm, an immunogen in nucleic acid-based vaccines [10,15,16,17]. Therefore, we speculated that some proteins, such as NSm, may influence specific immune responses of the host.

Previous studies have shown that NSm decreases host protection efficiency in the development of RVFV DNA vaccines [18]. Additionally, researchers have found that human pathogenic viruses usually possess antagonists of innate immunity in their NSs and NSm proteins in the *Bunyavirales* order. Although the function of CCHFV NSm is not completely characterized [27,36], it is known that NSm could improve Gc processing and secretion [27]. Thus, NSm should theoretically promote the immunogenicity of glycoprotein-based vaccines. However, nucleic acid-based vaccines, where the GPC immunogen includes NSm, do not provide 100% protection [10,16,37]. In contrast, the protection efficiency of GcGn mRNA vaccination was 100% in IFNAR^−/−^ mice [9]. To explore the role of NSm in CCHFV vaccination, we designed three mRNA candidates: vLMn, vLMc, and vLMs.

The LNP delivery system play a key role in mRNA expression efficiency. DLin-MC3-DMA has been evaluated for nucleic acid delivery in clinical trials, becoming a common benchmark for in vivo mRNA delivery [21,38]. Meanwhile, two formulations of LNPs encapsulating mRNA, BNT162b2, and mRNA-1273 have been successful for the prevention of severe COVID [39]. We used orthogonal experimental design and complete randomized design to optimize LNPs for the delivery of mRNA vaccines. Afterwards, we screened the most optimized LNP formulation by evaluating the LNP size, encapsulated efficiency, cytotoxicity, and expression efficiency in vitro and in vivo. Then, we generated three mRNA vaccines, and evaluated the B-/T-cell immune responses induced via vLMn, vLMc, vLMs, and mock vaccines in immunocompetent mice.

A previous study assessed immune responses induced by subunit vaccines encoding ectodomains Gn (eGn) and ectodomains Gc (eGc). They reported that no specific IgG antibodies were observed after initial vaccination with either eGn or eGc, but titers increased with subsequent immunizations [19]. In our studies, although prime immunization with vLMn did not induce specific IgG antibodies to Gn, IgG antibodies generated by vLMc against Gc were observed (Figure 6). This dissimilarity might be due to the mRNA vaccine having a higher immunogenicity. The antibody titer was also positively correlated with the number of immunizations from our results. In addition, another study demonstrated that the titer of IgG antibodies induced by pVAX-LAMP1-CCHFV-Gn was less than that of IgG antibodies induced by pVAX-LAMP1-CCHFV-Gc [40]. The B-cell immune responses generated through vLMn, vLMc, or vLMs also present the same conclusion. Interestingly, although we inferred that vLMs including NSm could impact the resulting immune response, it was surprising that vLMs induced lower levels of antibodies targeting both Gn and Gc, compared to those of vLMn or vLMc. A potential explanation is that the quantity of the expression of both Gn and Gc was decreased when using Cap1-mRNA-GnNSmGc transfected into 293T cells, which could have resulted in decreased immunogenicity (Figure 3C,D). Similar observations were made with DNA vaccines expressing both NSm and GnGc against RVFV [18]. Second, NSm may have influenced immune responses via a yet-to-be-characterized mechanism. Due to the lack of recombinant purified Gn or Gc proteins, the results of the ELISA assay are not sufficiently sensitive to detect IgM antibodies.

T cells and IFN-γ have been reported to play a key role in survival following a CCHFV infection [32,41]. The current data from animal studies show that using an mRNA vaccine platform expressing GcGn induced IFN-γ against Gc peptides. However, it did not activate IFN-γ against Gn peptides [9,10], which is consistent with our finding that Gc is a stronger immunogen compared to Gn in terms of IFN-γ activation (Figure 7). It has been reported that no T-cell epitopes are located within the Gn [42], which might explain why IFN-γ responses cannot be induced with Gn as an immunogen. Although Gn cannot elicit strong immune responses for IgG antibodies and IFN-γ, the maturation process of Gc depends on the Gn protein for correct localization, folding, and transport [43]. Also, Gn has B-cell epitopes [44], which indicates that Gn is still necessary for the development of CCHFV vaccines. In addition, a study found that the NP expressed by the mRNA vaccine generated an IL-4-biased response, compared to the response of IFN-γ [12]. Interestingly, all three mRNA vaccines could induce IL-4 responses (Appendix A). According to the results of IFN-γ and IL-4 (Figure 7 and Appendix A), more IFN-γ-specific spots were observed for the vLMc and vLMs groups, which demonstrated that these mRNA vaccines trend towards a Th1-biased immune response and have a potential ability to provide a protective effect against CCHFV [31]. As for the role of NSm in the T-cell response, the level of IFN-γ activated by vLMc against the Gc1 peptide was higher than that of vLMs to the Gc1 peptide (Figure 7), which is the same result as with the B-cell response. This may be explained by the possibility that NSm played a role in antagonizing the overall host immune responses [45].

Pre-clinical and clinical studies have previously shown that Gn, Gc, and NP have several linear B-cell epitopes [44,46,47] and that Gc and NP have the potential to induce T-cell immune responses [42,48]. Furthermore, the lack of serum antibodies usually leads to patient death [49,50], and neutralizing antibodies could often be detected during the convalescent phase in human survivors of CCHF [51]. Our results support the use of Gc-dominated and Gn-supplemented vaccine immunogens in the future and hint to avoid using the full M segment including NSm.

## 5. Conclusions

In conclusion, our studies demonstrated the adverse effects that the inclusion of NSm may have on the development of immunogenic CCHFV vaccines, while providing knowledge for the design of improved antigens in the future.

## Figures and Tables

**Figure 1 viruses-16-00378-f001:**
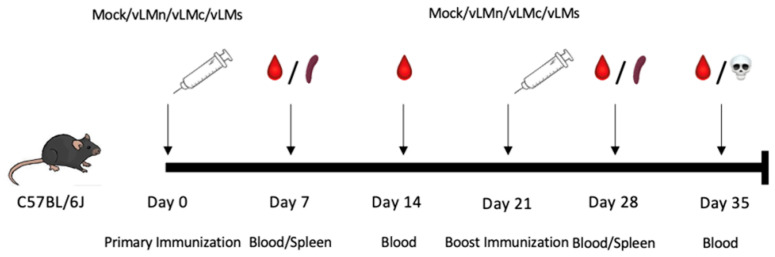
Schematic of the timeline for animal experiments. Vaccines (vLMn, vLMc, and vLMs) and mock vaccine (LNPs-Cap1-mRNA-Luc) were used to immunize C57BL/6J mice via i.m. at day 0 and day 21. Blood samples were collected at days 7 and 14 (7 and 14 days after the primary immunization) and at days 28 and 35 (7 and 14 days after the boost immunization) for ELISA assay. Spleen samples were collected at day 7 (7 days after the primary immunization) and at day 28 (7 days after the boost immunization) for ELISPOT assay.

**Figure 2 viruses-16-00378-f002:**
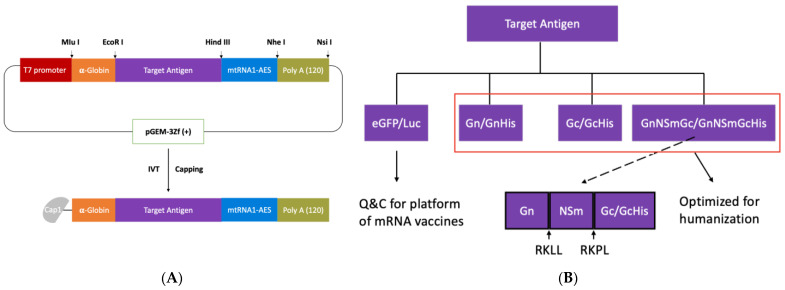
Genomic structure of mRNA template. (**A**) Corresponding Cap1-mRNA were synthesized by IVT and capping using constructed pGEM-3Zf (+) mt including target antigen as templates; 5′UTR was α-globin, 3′UTR was mtRNA1-AES, and poly A tail included 120 A. The short arrows indicate enzyme digestion. (**B**) Target antigens were eGFP, Luc, Gn, GnHis, Gc, GcHis, GnNSmGc, and GnNSmGcHis. The red box indicates immunogen genes encoding CCHFV glycoprotein that were only codon-optimized for humans. The dotted line indicates that GnNSmGc and GnNSmGcHis could be digested to Gn, NSm, and Gc/GcHis by SKI-1/S1P and SKI-1/S1P-like enzymes. The short arrows indicate sites for enzyme digestion.

**Figure 3 viruses-16-00378-f003:**
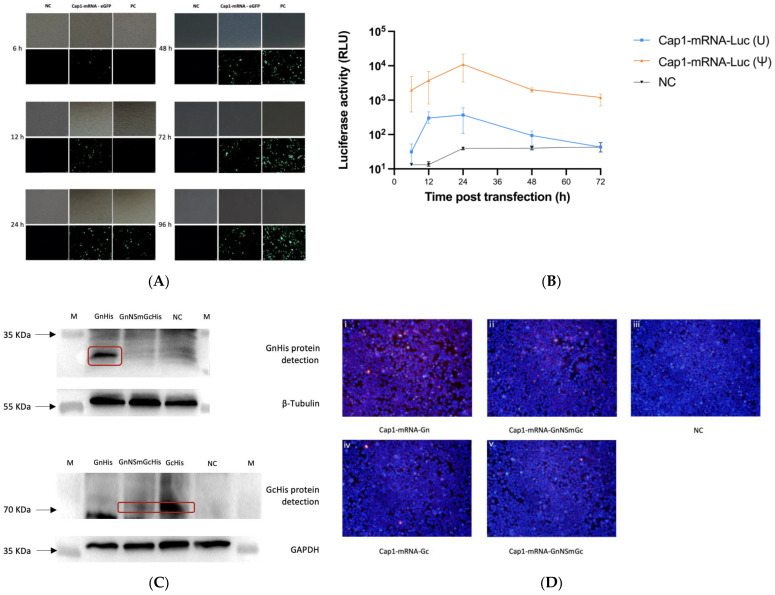
Protein expression assays. (**A**) Cap1-mRNA-eGFP was transfected into 293T cells and observed at 6 h, 12 h, 24 h, 48 h, 72 h, and 96 h to verify antigen expression compared with a negative control (NC) Cap1-mRNA and a positive control (PC) pVAX-eGFP; (**B**) modified Cap1-mRNA-Luc, unmodified Cap1-mRNA-Luc, and NC Cap1-mRNA were transfected into 293T cells and observed at 6 h, 12 h, 24 h, 48 h, and 72 h. RLU indicates relative light units. The data are shown as mean ± SD. (**C**) Expression of His-tagged GnHis/GcHis/GnNSmGcHis proteins in 293T cells were detected by Western Blot assay. M indicates marker. Arrows indicate marker location. The red box indicates expression location of target proteins; (**D**) expression of Gn (**i**,**ii**), eGFP (**iii**), and Gc (**iv**,**v**) proteins in 293T cells were detected with mice sera by immunofluorescence assay. Cap1-mRNA-eGFP as NC. Blue indicates DAPI staining, red indicates target proteins.

**Figure 4 viruses-16-00378-f004:**
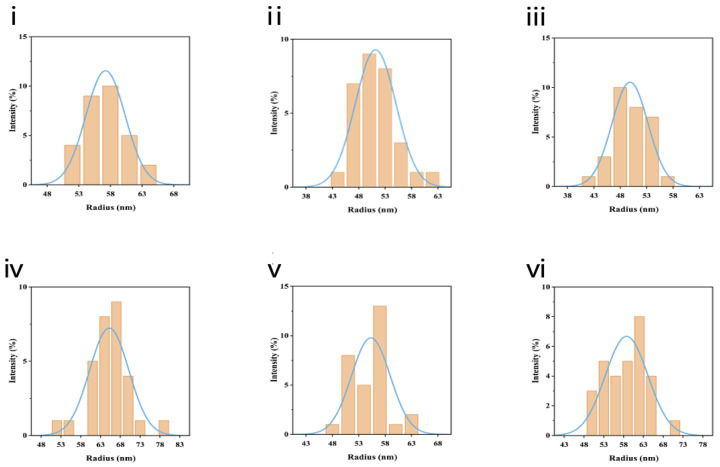
Intensity size graph of LNPs-Cap1-mRNA measured by dynamic light-scattering method. The physical characteristics of LNPs generated by methods 1, 2, 3, 4, 5, and 6 in Table 3 correspond to panels (**i**), (**ii**), (**iii**), (**iv**), (**v**), and (**vi**), respectively.

**Figure 5 viruses-16-00378-f005:**
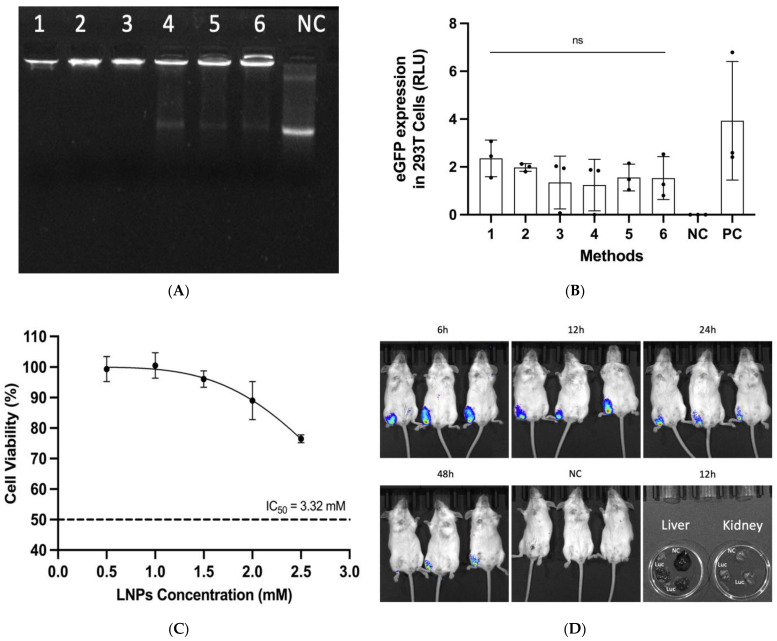
(**A**) Gel retardation assay. LNPs-Cap1-mRNA manufactured by methods 1–6 in Table 3 were run on 1.2% nuclease-free agarose gel. NC indicates negative control using naked Cap1-mRNA; (**B**) detection of expression intensity of LNPs-Cap1-mRNA-eGFP manufactured via methods 1–6 in 293T cells; naked Cap1-mRNA-eGFP as NC. Cap1-mRNA-eGFP encapsulated by LipofectamineTM 2000 Reagent as PC. Data are shown as mean ± SD, and there are no significant differences between different methods. (**C**) Cytotoxicity of LNPs-Cap1-mRNA was detected on 293T cells. The dotted line indicates a deductive IC_50_ of 3.32 mM. (**D**) In vivo bioluminescence image in mice. Female BALB/c mice were administered 10 μg of LNPs-Cap1-mRNA-Luc via the i.m. route and observed at 6 h, 12 h, 24 h, and 48 h post-vaccination. The liver and kidneys were observed at 12 h post-vaccination. The same volume of PBS via the i.m. route was used for the NC group.

**Figure 6 viruses-16-00378-f006:**
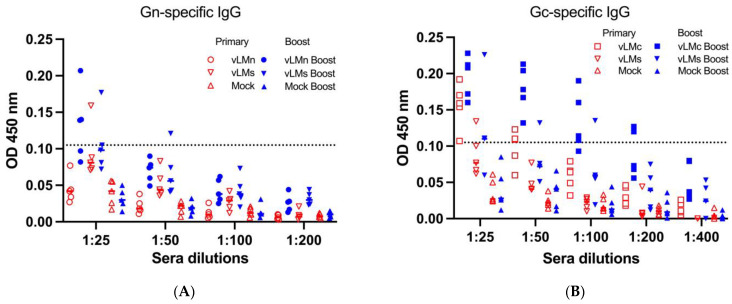
Specific IgG antibodies titer in C57BL/6J due to vaccination by mRNA vaccines. (**A**) Sera were collected from the vLMn and vLMs vaccine groups after primary and boost immunization, and the antibody titers of IgG targeting Gn were detected; (**B**) Sera were collected from vLMc and vLMs vaccine groups after primary and boost immunizations, and the titer of IgG antibodies targeting Gc was detected. The dotted lines represent the limit of detection.

**Figure 7 viruses-16-00378-f007:**
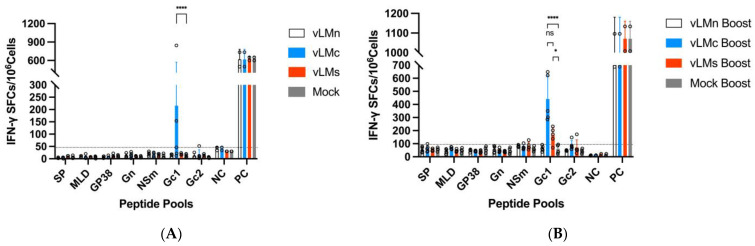
IFN-γ responses after prime and boost mRNA vaccination. Fresh splenocytes from mice were stimulated with spanning peptide pools including SP, MLD, GP38, Gn, NSm, Gc1, and Gc2 from GPC of CCHFV YL16070 strain. ELISPOT was used to determine the number of IFN-γ spot-forming cells (SFCs) per 10^6^ splenocytes (**A**) in prime and (**B**) boost vaccinations of vLMn, vLMc, vLMs, and mock. Medium was used as NC. PMA + Ionomycin was used as PC. Dashed lines indicate limit of detection. The symbols represent the individual animals in the groups. Data are shown as means ± SD and analyzed using two-way ANOVA with multiple comparison test. **** *p* < 0.0001, * *p* < 0.05, ns means not significant.

**Table 1 viruses-16-00378-t001:** The reaction system for in vitro transcription.

Name	Volume
Template ^1^	X μL ^2^
ATP (100 mM)	3 μL
ψ/UTP (100 mM)	3 μL
GTP (100 mM)	3 μL
CTP (100 mM)	3 μL
10 × Reaction Buffer	3 μL
T7 RNA Polymerase Mix	2 μL
Nuclease-free water	Add to 30 μL
Total reaction volume	30 μL

^1^ Linearized pGEM-3Zf (+) mt-eGFP/Luc/Gn/Gc/GnNSmGc/GnHis/GcHis/GnNSmGcHis. ^2.^ The volume of X is converted through microgram formulation, which is target sequence bp × 0.6.

**Table 2 viruses-16-00378-t002:** Factors and levels of the orthogonal experimental design.

Number	Molar Ratios (A) ^1^	Concentration (B) ^1^	pH (C) ^1^
1	45:40:13.5:1.5	10 mM	4
2	50:38.5:10:1.5	50 mM	5
3	46.3:42.7:9.4:1.6	75 mM	6

^1^ Brackets indicate factor name.

**Table 3 viruses-16-00378-t003:** Orthogonal experimental design.

Number	A ^2^	B ^2^	C ^2^
1	1 ^1^	1 ^1^	1 ^1^
2	1 ^1^	2 ^1^	3 ^1^
3	1 ^1^	3 ^1^	2 ^1^
4	2 ^1^	1 ^1^	3 ^1^
5	2 ^1^	2 ^1^	2 ^1^
6	2 ^1^	3 ^1^	1 ^1^
7	3 ^1^	1 ^1^	2 ^1^
8	3 ^1^	2 ^1^	1 ^1^
9	3 ^1^	3 ^1^	3 ^1^

^1^ Numbers correspond to the numbers in Table 3. ^2^ Letters correspond to factor names in Table 3.

**Table 4 viruses-16-00378-t004:** Complete randomized design.

Number	Molar Ratios	pH
1	45:40:13.5:1.5	4
2	50:38.5:10:1.5	4
3	46.3:42.7:9.4:1.6	4
4	45:40:13.5:1.5	5
5	50:38.5:10:1.5	5
6	46.3:42.7:9.4:1.6	5

## Data Availability

All available data are presented in the article.

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
