# Peer review of "A mRNA Vaccine for Crimean–Congo Hemorrhagic Fever Virus Expressing Non-Fusion GnGc Using NSm Linker Elicits Unexpected Immune Responses in Mice"

_viruses, 2024, doi:10.3390/v16030378_

Round 1

Reviewer 1 Report

Comments and Suggestions for Authors

The authors describe the development of mRNA vaccines for CCHFV encoding the Gn, Gc and GnNSmGc and subsequent characterization of immune responses generated by the vaccines in an animal model.  Clearly, the Gc based vaccine induced superior immune responses.

Minor comments

Line 26: Crimean not Crimeans

mRNA synthesis: What test(s) were done to confirm complete DNA elimination following DNase treatment?

Line 236: Secondary antibodies for ELISA should be stated.

The authors should state that the mRNA vaccines encode full-length Gn and Gc.  

Figure 3Dii. Not clear what protein expression is being shown (Gn or Gc or Both). Line 294 should indicate Gn expression is the focus.  

Figure 3D: The images are not clear.

Expression time and intensity of LNPs-Cap1-mRNA-Luc in mice. Did the authors check if the LNPs did spread to the rest of the mouse body, or they remain localized to injection site? This may have implications on the safety of the vaccines delivered by LNPs.

Line 481: Need to specify that this is in animal studies.

Comments on the Quality of English Language

The article needs editing of the English language.  There are some grammatically incorrect sentences in the article. 

Reviewer 2 Report

Comments and Suggestions for Authors

The study describes the immune responses to RNA vaccines containing CCHFV Gn, Gc or GnNSmGc as vaccine antigens. The study also compares immune responses to Gn and Gc when NSm is included in the vaccine construct to see the negative effect that might be exerted by NSm. While the study title and abstract sounds interesting, there are concerns about the results, and experimental design. The details of these concerns, together with some edits and suggestions for each section and for some of the figures, are described below.

INTRODUCTION

1.     Line 26: Replace "Crimeans" with "Crimean."

2.     Line 42, only one nonstructural protein from the S segment has been identified so far, nonstructural protein instead of nonstructural proteins.

3.     Ref 9, the authors could directly refer to the corresponding WHO webpage instead of other papers referring to WHO.

Even though more studies are mentioned later in the manuscript, consider referring to additional papers between Line 45-52. Include GP38 in the brief description of CCHFV antigens explored as vaccine antigens.

RESULTS

1.     Based on the schematic representation, spleen collection was not performed at days 14 and 35. The legend of Figure 1 should be rephrased accordingly.

2.     Is it possible to give more details on the constructs generated to express Gn, Gc and GnNSmGc?  

3.     The authors chose to use anti-His secondary antibody for the Western blots, while they used Gn/Gc serum for immunofluorescence staining. This Western blot analysis was not able to detect Gn in Gn/NSm/GcHis transfected cells, and we can not compare the expression level of Gn with Western blotting.

4.     The bands from GcHis Western blots also are not clear.

5.     The immunofluorescence assay shows the Gn expression with the GnHis construct, and the Gn levels are lower in the GnNSmGcHis construct. If this study is comparing immune responses to individual proteins in different vaccine constructs and the effect of having NSm on immune responses to Gn and Gc proteins, the expression levels of these proteins should be similar. Together with the unclear detection of the bands corresponding to Gc with Western Blot, the immunofluorescence assay also shows only a few cells expressing Gc. Similar concerns also stand for the expression levels of Gc from the GnNSmGc construct. Different anti-Gn and anti-Gc antibodies could be used to improve Western blot and IFA.

6.     Line 360, humoral.

7.     The experiment timeline shows multiple time points for blood collection and the time points of the post-prime and boost samples that have been used to detect antibody titers were not clear both in the manuscript or in the figure legend. Is it D14 for prime and D35 for boost?

8.     The authors state that Gn and Gc-specific antibody titers were lower when the vaccine construct included NSm, yet no statistical analysis was performed to show if these differences are reaching significance. Would it be possible to give data of individual animals? If all the animals in vaccine groups do not have similar antibody titers, showing titers of individual animals could be informative.

9.     Could the low antibody titers be the result of assay sensitivity? What is the expression level of Gn and Gc in 293T cells? A Western blot analysis could be performed to show the expression levels of these proteins. Are these sequences from the same CCHFV strain?

10.  Line 382, figure legend, antibody titers.

11.  The manuscript states the ELISPOT assays were performed with pooled splenocytes, and the graphs have five symbols. Do the symbols on the graphs represent the individual animals in the groups or the experimental replicates?

12.  Line 404, GPC instead of GP.

13.  The segments of the graphs both in Figure 7B and 8B could have been adjusted to allow full visibility to the bars representing results from vaccine groups.

14.  The lines and asterisks showing the pairwise comparisons are far from the bars, making it harder to see which bars are being compared. Is the difference between the recall responses to Gc1 pool of vLMc and vLMs vaccinated animals statistically significant?

15.  Figure 8B shows comparable numbers of IL-4-secreting splenocytes from vLMc and vLMs vaccine groups to the NSm peptide pool. The authors also state in the text that “The IL-4 response to NSm peptide was enhanced post boost with both vLMc and vLMs.” This was confusing and needs to be re-evaluated since the vLMc vaccine construct is just Gc, as it has been stated in the discussion, and does not contain NSm.

16.  Similarly, Figure 8A has confusing comparisons. The figure shows a significant increase in the IL-4-secreting cell numbers to GP38, NSm, Gc1, and Gc2 peptide pools in animals vaccinated with vLMn (Gn). Does the vLMn construct contain proteins other than Gn? Authors should address these discrepancies.

17.  The ELISPOT graph could be plotted to allow more visualization of experiment groups. One suggestion could be plotting the positive controls in a different graph. This would allow the authors to adjust the Y-axis accordingly.

DISCUSSION

1.     Lines 427-429: Would it be okay to say CCHFV NP is less immunogenic based on one vaccine paper? There are vaccine studies showing complete protection from the lethal infection when NP is used as the vaccine antigen. Vaccine platform-based differences in immune responses and the level of conferred protection should be kept in mind.

2.     Lines 429-430: There are vaccine studies with CCHFV that show that neutralizing antibodies are not required for protection from lethal infection (PMID: 28250124, PMID: 33654101).

3.     Line 443: Only reference 18 is a vaccine study.

4.     The authors refer to vaccine papers that have CCHFV GPC and do not confer protection or confer partial protection and speculate that having NSm is the reason, yet there are vaccine studies with CCHFV GPC that confer complete protection. This speculation disregards vaccine-platform-based differences in immune responses.

5.     Based on their results, the authors conclude that the inclusion of NSm may have adverse effects on the immunogenicity of the CCHFV vaccines, yet the expression levels of Gn and Gc were decreased in the construct that has NSm, which makes comparison of immune responses not possible since similar amount of antigens were not introduced to the immune system. Authors should address these limitations.

Reviewer 3 Report

Comments and Suggestions for Authors

The manuscript by Chen et al describes optimisation of a delivery system for an mRNA vaccine against Crimean-Congo hemorrhagic fever virus (CCHFV), and characterises the responses to different vaccine constructs in a mouse model. This is an interesting report, with comprehensively described methods. Whilst there are some inherent limitations in the methods, the authors are transparent about these.

The study presented is complex and as such the figures are a key aspect of the manuscript. The rendering of figures 2A, 2D, 4 is small and as such some of the detail difficult to discern - larger and / or higher resolution figures here would improve the manuscript. Elsewhere, enhancing annotations on figures (e.g. annotating figure 5A, integrating antigen target of figure 6 A & B into the figure etc.) would improve quality.

The authors should proof the manuscript as described below, and additionally ensure the use of terms for the disease (Crimean-Congo hemorrhagic fever) and the virus (Crimean-Congo hemorrhagic fever virus) are correctly applied throughout. 

Comments on the Quality of English Language

Overall the quality of English is good, but with the complexity of methods and data presented, is densely written in places and would benefit from proofreading to help facilitate the readability of the paper. Further, there are spelling errors intermittently in the main manuscript and figure legends which should be addressed.

Round 2

Reviewer 2 Report

Comments and Suggestions for Authors

The study describes the immune responses following the vaccination with mRNA vaccines containing CCHFV Gn, Gc, and NSm. The authors investigated the humoral immune responses following vaccination as well as T cell responses through the investigation of IFN-gamma recall responses. Both antibody responses and the number of IFN-gamma-secreting cells have higher immune responses with mRNA vaccine with the construct containing Gc alone. The manuscript provides information on how using different antigens in various vaccine platforms can result in different outcomes, ultimately affecting the immune responses, compared to other published studies.

I would like to thank the authors for considering the edits and suggestions made throughout this reviewing process and for their efforts in corresponding. Even though the study has limitations, these limitations were addressed in the manuscript. My only comment will be on the study referred to on Line 55, which is an inactivated rhabdoviral vaccine study.

Author Response

Sorry for the internet freeze, I don't know if the attachment upload, so I upload it again. Please see the attachment.
